# Towards Home-Based Diabetic Foot Ulcer Monitoring: A Systematic Review

**DOI:** 10.3390/s23073618

**Published:** 2023-03-30

**Authors:** Arturas Kairys, Renata Pauliukiene, Vidas Raudonis, Jonas Ceponis

**Affiliations:** 1Automation Department, Electrical and Electronics Faculty, Kaunas University of Technology, 51368 Kaunas, Lithuania; 2Department of Endocrinology, Lithuanian University of Health Sciences, 50161 Kaunas, Lithuania; 3Institute of Endocrinology, Lithuanian University of Health Sciences, 44307 Kaunas, Lithuania

**Keywords:** diabetic foot ulcer (DFU), burn, chronic wounds, deep learning, convolutional neural network (CNN), datasets

## Abstract

It is considered that 1 in 10 adults worldwide have diabetes. Diabetic foot ulcers are some of the most common complications of diabetes, and they are associated with a high risk of lower-limb amputation and, as a result, reduced life expectancy. Timely detection and periodic ulcer monitoring can considerably decrease amputation rates. Recent research has demonstrated that computer vision can be used to identify foot ulcers and perform non-contact telemetry by using ulcer and tissue area segmentation. However, the applications are limited to controlled lighting conditions, and expert knowledge is required for dataset annotation. This paper reviews the latest publications on the use of artificial intelligence for ulcer area detection and segmentation. The PRISMA methodology was used to search for and select articles, and the selected articles were reviewed to collect quantitative and qualitative data. Qualitative data were used to describe the methodologies used in individual studies, while quantitative data were used for generalization in terms of dataset preparation and feature extraction. Publicly available datasets were accounted for, and methods for preprocessing, augmentation, and feature extraction were evaluated. It was concluded that public datasets can be used to form a bigger, more diverse datasets, and the prospects of wider image preprocessing and the adoption of augmentation require further research.

## 1. Introduction

Diabetes mellitus is a metabolic disorder resulting from a defect in insulin secretion, resistance, or both [1]. According to a 2021 estimate made by International Diabetes Federation, 537 million adults, or 1 in 10 adults worldwide, had diabetes, and an additional 541 million adults were at high risk of diabetes [2]. Diabetes results in high blood glucose levels, among other complications; these can damage blood vessels and nerves and may result in peripheral neuropathy and vascular disease. These two conditions are associated with foot ulcer development. It is estimated that diabetic foot ulcers (DFUs) will develop in up to 25% of diabetes patients [3]. Patients with DFUs have a high risk of amputation, and this was found to result in mortality over 5 years for 50% of patients [4]. Periodic checks help prevent lower-limb amputations and even foot ulceration, but this creates a large burden for the healthcare system. This burden can be reduced by employing computer vision technologies based on deep learning (DL) for ulcer diagnosis and monitoring without increasing the need for human resources.

For the ImageNet dataset [5], the creation and adoption of graphics processing units (GPUs) led to the rapid development of deep learning algorithms and their application, including in the field of computer vision. Computer vision has already been used in medical imaging for multiple decades, but the adoption of deep learning technologies has fostered its application for multi-modal image processing in other more complex medical domains [6]. Computer vision technologies based on machine learning, including deep learning, have been applied for wound assessment, and multiple reviews have already been conducted. Tulloch et al. [7] found that the most popular research was related to image classification, and the reported results were only limited by the lack of training data and the labeling accuracy. Additionally, it was concluded that the application of thermography is limited by the inability to use asymmetric detection and the high equipment costs. A review dominated by shallow machine learning solutions concluded that 3D (three-dimensional) imagery and deep learning methods should be used to develop accurate pressure ulcer assessment tools [8], but the costs related to 3D imaging were not considered. Chan and Lo [9] performed a review of existing DFU monitoring systems, with some using multiple image modalities, to conclude that, while they are useful in improving clinical care, majority have not been reviewed to prove wound measurement accuracy. A review of ulcer measurement and diagnosis methods performed by Anisuzzaman et al. [10] concluded that datasets of multimodal data types are needed in order to apply artificial intelligence (AI) methods for clinical wound analysis. Another review of state-of-the-art (SOTA) deep learning methods for diabetic foot ulcers evaluated performance in terms of three main objectives: classification (wound, infection, and ischemia), DFU detection, and semantic segmentation. Small datasets and a lack of annotated data were named as the main challenges for the successful adoption of deep learning for DFU image analysis [11]. Similar conclusions were drawn in a wider wound-related review [12] in which different wound types were considered.

Multiple past reviews concluded that multimodal data are able to deliver better classification, detection, and semantic segmentation results in the clinical care of wounds. The lack of annotated training data was identified as the main challenge in SOTA deep learning applications. This paper seeks to review the latest applications of deep learning in diabetic foot ulcer image analysis that are applicable in the context of a home environment and can be used with a conventional mobile phone. The findings of this review will be used in the development of a home-based DFU monitoring system. It is inevitable that some of the findings overlap with those of existing reviews. This paper seeks to demonstrate that:Research in this field can benefit from publicly available datasets.The current research has not fully facilitated data preprocessing and augmentation.

As a result, the current review has the following contributions:Publicly available wound datasets are accounted for.The preprocessing techniques used in the latest research are reviewed.The usage and techniques of data augmentation are analyzed.Further research avenues are discussed.

This article is organized as follows. Section 2 presents the methodology used for search for and selection of scientific articles. The selected articles are reviewed in terms of their performance and methodology in Section 3. Section 4 lists the limitations of this review, and the article is concluded with a discussion on the findings in Section 5.

## 2. Methods

The search for and selection of articles were conducted by using the Preferred Reporting Items for Systematic Reviews and Meta-Analyses (PRISMA) methodology [13]. The current section will explain the article search and selection process in detail.

### 2.1. Eligibility Criteria

Papers were selected by using the eligibility criteria defined in Table 1. Articles’ quality, availability, comparability, and methodological clarity were the key attributes considered during the definition of the eligibility criteria. The criteria included the following metrics: publishing language, paper type, full paper availability, medical domain, objective, data modality, usage of deep learning, and performance evaluation. Initially, exclusion criteria were used to filter out articles based on their titles, abstracts, and keywords. The remaining papers were read fully and further evaluated for eligibility.

### 2.2. Article Search Process

The articles reviewed in this paper were collected by using two search strategies: a primary and a secondary search. The primary search was performed by using the following electronic databases:Web of ScienceScopus.

The database search was conducted on 17 December 2022, and articles that had been published since 2018 were considered. This time range limited the number of search results that were returned and helped ensure that the selected articles used the latest available deep learning methods and dataset preparation protocols. The search queries contained the following information categories:Terms for wound origin (diabetic foot, pressure, varicose, burn);Terms for wounds (ulcer, wound, lesion);Terms for objectives (classification, detection, segmentation, monitoring, measuring);Terms for neural networks (artificial, convolutional, deep).

Boolean logic was used to define queries by joining term categories with a logical AND gate and concatenating terms with a logical OR gate; popular abbreviations (e.g., DFU for diabetic foot ulcer, CNN for convolutional neural network, etc.) were included. Some terms were truncated to add wildcard symbols in order to extend the search results (e.g., classif*, monitor*, etc.). During the search, queries were only applied on titles and abstracts.

During the secondary search, the references found in articles from the primary search were reviewed and manually included based on their relevance according to the eligibility criteria. Similar review articles [7,8,9,10,11,12] were additionally consulted for references, and these references were also included in the secondary search pool.

### 2.3. Selection Process

The selection was performed by the main author. The results from the primary search were imported into Microsoft Excel by exporting a CSV file from the search databases. The imported information included the title, authors, publication date, article type, where the article was published, the full abstract, and keywords. Duplicates were removed based on the title by using a standard Excel function. Non-journal and non-English articles were filtered out. The titles, abstracts, and keywords of the remaining articles were screened by using the criteria presented in Table 1. The remaining articles were read fully to exclude those that did not meet the set criteria. Forward snowballing [14] and a review of additional reference lists in other review articles resulted in an additional seventeen articles that passed the eligibility verification and were added to the final article set. The overall article selection procedure is presented in a PRISMA chart in Figure 1.

### 2.4. Data Extraction

The aims that were set for this review dictated the data type to be extracted. The online databases provided some metadata related to the articles, but in order to quantitatively evaluate them, the following data were extracted during full reading of the articles: the objective, subject, classes, methodology, initial sample count, training sample count, preprocessing and augmentation techniques, and deep learning methods that were applied. To avoid missing important information, the data extraction was independently performed by the first two authors. Unfortunately, not all articles specified the required data, and as a result, the following assumptions about the missing information were made:Training sample count—the initial sample count was considered for the evaluation of the training dataset size, and the training sample count is not listed here.Preprocessing technique usage—it was considered that no preprocessing techniques were used.Augmentation technique usage—it was considered that no augmentation techniques were used.

### 2.5. Data Synthesis and Analysis

For a performance comparison, the collected data were split based on the objective into three categories: classification, object detection, and semantic segmentation. Then, the data were ordered based on the publication year and author. This segregation was used to provide an overview of all collected articles and to describe the best articles in terms of performance metrics.

Analyses of the datasets, preprocessing and augmentation techniques, and backbone popularity and usage were conducted on the whole set of articles.

## 3. Results

The search in the Scopus and Web of Science databases returned 1343 articles. The articles’ titles were checked to remove duplicates, which resulted in 891 original articles. This set had titles, abstracts, and keywords that were evaluated against the eligibility criteria (Table 1). For the remaining 70 articles, the same criteria were used to evaluate their full texts. Finally, 35 articles were selected, and an additional 17 were added as a result of forwards snowballing and a review of the references in similar review papers. The resulting set of articles was further reviewed to extract the quantitative and qualitative data that were analyzed. This section provides the results of an analysis of the selected articles.

The distribution of articles based on the date and objective is provided in Figure 2. It can be observed that interest in this research field has been increasing for the last 5 years (with the exception of 2021). The distribution of objectives shows that classification and semantic segmentation were researched the most.

### 3.1. Performance Metrics

Multiple standard performance metrics are used for model evaluation and for benchmarking against the work of other authors. This section provides details about the most commonly used metrics and their derivations. Metrics that were used by more than five (inclusive) articles will be described; only these measurements are presented in tables listing the reviewed articles.

Model performance is quantified by applying a threshold on model prediction. For classification, a threshold is directly applied to the model output. For object detection and semantic segmentation, model prediction is evaluated by using the intersection over union (IoU), and a threshold is applied to the result. In object detection, the result is a bounding box that contains a region of interest (ROI), and for semantic segmentation, it is a group of image pixels that contain the same semantic information. As shown below, the IoU is calculated by dividing the number of image pixels (px) that overlap in the predicted area (X) and the ground-truth area (Y) by the number of image pixels that are in both areas X and Y.
(1)$$IoU=\frac{X\cap Y}{X\cup Y}=\frac{px_{TP}}{px_{TP}+px_{FP}+px_{FN}}$$

Thresholding results are used to create confusion matrix and are grouped into four self-explanatory categories: true positive (TP), true negative (TN), false positive (FP), and false negative (FN). Confusion matrices help evaluate a model’s performance and make better decisions for fine-tuning [15]. A confusion matrix is summarized by using the calculated metrics (see Table 2) to specify the model performance by using normalized units. It should be noted that some articles also used the mIoU performance measure, which is calculated as the mean of the IoU of all images.

Performance metrics were analyzed in all selected articles, and it was found that 94% of the articles could be compared by using the accuracy, F1, mAP, and MCC metrics. Though their calculation and meanings are not the same, their normalized form provides a means for comparison.

### 3.2. Datasets

Deep learning models eliminate the need for manual feature engineering, but this results in a high model parameter count. To train such models, large datasets are required; though they are general-purpose, public datasets have accumulated enough samples (e.g., ImageNet [5]), and model development is only limited by inference time constraints and the hardware on which the model will run. Publicly available medical data are not abundant, and this creates limitations for deep learning applications. Ethical implications limit the collection of medical images, and the use of existing imagery is limited by a lack of retrospective patients’ consent. In this context, researchers are motivated to collaborate with medical institutions and other researchers to collect new samples or to reuse valid publicly available data.

All reviewed articles used images that represented three color channels: red, green, and blue (RGB). This was one of the conditions applied during article selection, and it makes the findings applicable in a home-based solution in which a conventional mobile phone can be used to take images. This section aims to account for all publicly available datasets by listing the datasets reported in the reviewed research.

Table 3 lists all of the datasets that have been made publicly available. The datasets were grouped based on their originating institution(s), and the “References” column provides the article(s) that presented the dataset. The “Sets Used” column indicates how many different datasets originated from the same institution, the “Use Count” column provides the total number of uses in the selected articles, and the “External Use References” column provides additional articles that referenced the listed dataset. Though the majority of the public datasets used contained diabetic foot ulcers, other dermatological conditions were also included to extend the available data.

The first three datasets are public, but were not used in any of the selected articles. The Medetec dataset [24] is an online resource of categorized wounds, and different studies have included it in different capacities; the table above provides the largest use case. Datasets 7 and 10 are used the most, but at the same time, they originated in multiple articles from a single medical/academic institution. This creates a risk that duplicate images are included in the total number; therefore, a duplicate check should be applied prior to using these datasets. Some of the research that was performed with dataset 10 included image patch classification, but the patches were not included in the count to avoid misleading estimation.

Though the combined number of public images was 11,173, the three largest proprietary datasets used contained 482,187 [48], 13,000 [49], and 8412 [50] images prior to applying any augmentation. Most of the public datasets had a simple image collection protocol (constant distance, conventional illumination, and perpendicular angle), and though samples within a single set were collected under constant conditions, using images from different datasets can be beneficial in improving dataset diversity. Different patient ethnicities, illumination conditions, and capturing devices create diversity, which is beneficial when training a model to be used at home by non-professionals with commercially available smartphones.

### 3.3. Preprocessing

Data quality determines a CNN model’s capability to converge during training and generalize during inference. Dataset quality is affected by the protocol used during image capture, the cameras, lenses, and filters used, the time of day, etc. All of these factors introduce variability that can be addressed before model training with a technique called preprocessing. The context of home-based monitoring increases the importance of preprocessing because different image-capturing devices will be used, and the illumination guidelines will be difficult to fulfill.

As seen in Figure 3, the majority of the selected articles used one or no (or unreported) preprocessing techniques. Here, all articles that used public or a mix of public and proprietary datasets were assigned to the **Public** class, and the **Proprietary** class only included research in which publicly available data were not used.

The most commonly used techniques were cropping (mentioned in 17 articles) and resizing (mentioned in 15 articles). Cropping was used to remove background scenery that could affect the image processing results. Resizing was used to unify a dataset when different devices were used to capture images or when cropping resulted in multiple different image sizes. Three articles reported [32,50,51] the use of zero padding instead of resizing to avoid the distortion of image features. Non-uniform illumination was addressed by applying equalization [17], normalization [51,52,53], standardization [22], and other illumination adjustment techniques [16]. Color information improvement was performed by using saturation, gamma correction [17], and other enhancement techniques [54]. Sharpening [17,53], local binary patterns (LBPs) [46], and contrast-limited adaptive histogram equalization (CLAHE) [16,55] techniques were used for image contrast adjustment. Noisy images were improved by using median filtering [56] or other noise removal techniques [16]. Images taken using a flash were processed to eliminate overexposed pixels [36,57]. Other techniques included grayscale image production [58], skin detection by using color channel thresholding [22], and simple linear iterative clustering (SLIC) [59,60].

Though multiple preprocessing techniques were reported, the motivations for their usage were not specified. Therefore, we can only speculate about the goal of their application. A handful of articles applied multiple techniques, but without the reporting of improvements, we can only predict that the inference time overhead was justified. At the same time, numerous articles did not report any preprocessing techniques, and this raises the question of whether the best result was actually achieved or if it was only a poorly presented methodology. One of the public datasets [40] contained augmented data that were preprocessed before augmentation. As a result, the articles that used this dataset did not used any preprocessing or augmentation techniques [17,42,43,44,45].

### 3.4. Data Augmentation

As it was found during the public dataset analysis that most of the selected research articles used a limited number of image samples. In order to train and test a model, a large number of samples is required, even when the model backbone is trained by using transfer learning. Data augmentation techniques can be employed to increase dataset size by using existing data samples. This is a cost-effective substitute for additional data collection and labeling, is well controlled, and was found in the past to overcome the problem of model overfitting [61].

The chart in Figure 4 shows the augmentation techniques that were used the most. Image flipping and rotation were used in most of the articles. While most of the articles performed all possible (vertical, horizontal, and both) flipping operations, some articles only used one type of flipping [52,62,63]. Only some of the articles specified rotation increments. In those, 25°, 45°, and 90° increment values were used, but this selection was not objectively verified, and the only obvious reasons could be balancing a multiclass dataset and avoiding too much data resulting from a single augmentation technique. Cropping and scaling techniques were used to vary wound proportions in images, to isolate wounds within images, and to change the amount of background information. Other techniques can be segmented into two large groups: geometric transformations (shift and shear) and color information adjustments (contrast, brightness, Gaussian blur, color space change, and color jittering and sharpening). The first group affects contextual information by changing a wound’s location or shape, while the second group changes images’ color features.

As seen in Figure 5, data augmentation was not related to the dataset source. Though most of the research did not use or report the use of augmentation, there were no general tendencies, and varying numbers of techniques were used. Most of the techniques were used in articles that produced public datasets [40], and this set a positive precedence when making data public. The authors invested time and effort in producing a balanced dataset that could encourage more reproducible research, and the results in different studies can, thus, be objectively compared. At the same time, it should be noted that even a single augmentation technique can produce multiples of an original dataset; e.g., Li [22] reported that they increased the training dataset by 60 times, but did not specify what augmentation techniques were used. Even two techniques would be enough to produce this multiplication, but here, concern should be expressed as to how useful such a dataset is and if a model trained with it does not overfit. Notably, Zhao [53] reported that image augmentation by flipping reduced the model’s performance, but this could also be an indication that model overfitting was reduced. One possible solution to this problem is training-time augmentation with randomly selected methods [41,52,64,65,66].

Based on the image augmentation taxonomy presented by Khalifa [61], almost all augmentation techniques that were found fell into the category of classical image augmentation and geometrics, photometrics, or other subcategories. Style-GAN [67] was the only deep learning data augmentation technique that was used for synthetic data generation [68]. According to Khalifa [61], this method falls into the deep learning augmentation subcategory of generative adversarial networks (GANs). The other two subcategories are neural style transfer (NST) [69] and meta-metric learning [70].

### 3.5. Backbone Architectures

In computer vision based on convolutional neural networks, an unofficial backbone term is used to describe the part of the network that is used for feature extraction [71]. Though this term is more commonly used to describe feature extraction in object detection and segmentation architectures, here, it will be used for all considered deep learning objectives, including classification.

The results in Figure 6 display most of the backbone architectures that were used in the reviewed articles. To achieve a better representation, some different versions of popular networks (e.g., MobileNet [72], Inception [73], VGG [74], and EfficientNet [75]) and custom-made architectures were grouped (see the two different bar colors).

The chart indicates that the small size of the datasets used in the reviewed articles motivated researchers to mostly rely on off-the-shelf feature extraction architectures that were pretrained on large datasets, such as ImageNet [5] and MS COCO [76]. ResNet [77] was found to be the most popular architecture; it was used in 42% of the analyzed articles. Feature extractors taken from classifiers of various depths (18, 50, 101, and 152 layers) were used for all three analyzed objectives, but in the majority of the articles (14 out of 22), they were used as a backbone for the network to perform semantic segmentation. Custom convolutional structures were found to be the second-largest subgroup of feature extraction networks (15 out of 52). There were two main motivations for using custom feature extractors: a lack of data and the aim of developing an efficient subject-specific network [51].

Custom CNNs were grouped into three groups based on their topologies: depth-wise networks, networks that used parallel convolution layers, and networks that used residual or skip connections. The use of a depth-wise topology is best explained when the image network size is considered. Two of the shallowest networks [36,57] were used to process 5 × 5 pixel image patches while using backbones with two and three convolutional layers. In other cases, the increase in the number of feature extractor layers matched the increase in the processed images’ resolution; four convolutional layers were used to extract features from 151 × 151 [46] and 300 × 300 [42] pixel images. Networks with parallel convolutions were inspired by the Inception [73] architecture and sought to extract features of multiple scales. All of them [17,51,78,79,80] were used to process image patches containing wounds in classification tasks. Networks within the last group were designed by using concepts of 3D [26] and parallel convolutions [30,47], repeatable network modules [44,81], and other concepts [66]. All of them used residual connections to promote deeper gradient backpropagation training, but less complex networks [77]. The use of two other publicly available backbones was motivated by their applicability for mobile applications (MobileNet) or multiscale feature detection (Inception).

Considering the different objectives, datasets, and annotations that were used, it was difficult to objectively compare the backbones’ performance, but here, we attempt to compare the effectiveness of feature extraction. Figure 7 shows an evaluation of the model performance while using the most common metrics, as discussed in Section 3.1; when multiple values were provided (in the case of multiclass classification performance), the best result was taken. The best, worst, and median performance data are shown.

Though all of the reviewed articles described their methodologies, for a few of the articles, it was difficult to objectively evaluate the backbones’ performance. Some studies presented binary classification results for multiple different classes [40,41,43,44,45,46], while others used unpopular performance metrics [22,23,58] and were excluded from this analysis. It could be observed that a performance metric of 0.9 or higher was achieved by using custom architectures, Inception, ResNet152, and other pretrained networks. ResNet architectures, which were used the most (with depths of 101 and 50), demonstrated high variability in performance, and the median value proved that this was a tendency and not the effect of an outlier. Conversely, the custom architectures showed a small variability, and this demonstrated their fine-tuning for the subject.

To objectively evaluate the backbones’ performance, the correlations between the performance and training datasets were calculated, as shown in Figure 8. The evaluation of these correlations was problematic because the augmentation results were not disclosed in all articles, and it was not clear what the actual training datasets’ sizes were. The same applies to articles that did not provide any details about augmentation. Correlations that did not reach extreme values of −1 or 1 indicated that the results were compared with those of similar dataset sizes. It can be concluded that the performance of all pretrained backbones improved when more training data were used. Network fine-tuning resulted in better generalization when more data were provided. A negative correlation for custom parallel and depth architectures showed that an increase in dataset size resulted in a performance reduction. Though the performance, as seen in Figure 9, was outstanding, the negative correlation indicated that more diverse datasets should be used to objectively verify the design.

### 3.6. Performance Comparison

The features extracted in a backbone network are processed to make a decision in a network element called a head. Different heads output different results and, thus, define a network objective. A classifier head will output the probability of an image belonging to two or more classes. A regression head that is used in object detection will output fitted bounding-box coordinates. A segmentation head will output an input image mask in which each pixel is individually assigned to a class. This subsection analyzes the architectures used in full and attempts to highlight the customizations that yielded the best performance. All reviewed articles were grouped by objective and ordered by date of publication and the main author. The best results for each objective are reviewed in detail after the presentation of a related table. Table 4 presents classification studies, Table 5 lists reviewed object detection articles and Table 6 provides details about semantic segmentation papers.

All classification articles were filtered to obtain those with F1 scores of at least 0.93 and training datasets with 9000 or more samples (at least 1000 non-augmented samples); six articles were found. Al-Garaawi et al. used local binary patterns to enrich visual image information for binary classification [46]. A lightweight depth CNN was proposed for the classification of DFUs, the presence of ischemia, and infections. Though the classification used achieved the poorest results (F1 score: 0.942 (wound), 0.99 (ischemia), and 0.744 (infection)), the smallest dataset was used for model training. Similar results were achieved (F1 score: 0.89 (wound), 0.93 (ischemia), and 0.76 (infection)) [43] by fusing features extracted by using deep (GoogLeNet [73]) and machine learning algorithms (Gabor [82] and HOG [83]) and applying a random forest classifier. It was demonstrated that fused features outperformed deep features alone. Though the same public datasets were used in both cases, here, augmentation was not used. The deeper and more complex (29 convolutions in total) architecture designed by Alzubaidi et al. [30] achieved an F1 score of 0.973 and was demonstrated to perform well in two different domains: diabetic foot ulcers and the analysis breast cancer cytology (accuracy of 0.932), outperforming other SOTA algorithms. Parallel convolution blocks were used to extract multiscale features by using four different kernels (1 × 1, 3 × 3, 5 × 5, and 7 × 7 pixels), a concept that is used in Inception architectures. Residual connections were used to foster backpropagation during training and solve the vanishing gradient problem. Das et al. proposed repeatable CNN blocks (stacked convolutions with the following kernel sizes: 1 × 1, 3 × 3, and 1 × 1 pixels) with residual connections to design different networks of different depths [44]. Two different networks of different depths were used; four residual blocks were used for ischemia classification and achieved an F1 score of 0.98, while a score of 0.8 was achieved in infection classification with a network containing seven residual blocks. This difference in classifier complexity and performance was explained by an imbalance in the training datasets, as the ischemia dataset had almost two times more training samples. Liu et al. used EfficientNet [75] architectures and transfer learning to solve the same problem of ischemia and infection classification. Though they used more balanced datasets, two different networks of different complexities were used to arrive at comparable performances. EfficientNetB1 was used for ischemia classification (F1 score: 0.99), while infections were classified by using EfficientNetB5 (F1 score: 0.98). Finally, Venkatesan et al. reported an F1 score of 1.0 in binary diabetic foot wound classification by using a lightweight (11 convolutional layers) NFU-Net [78] based on parallel convolutional layers. Each parallel layer contained two convolutions: feature pooling (using a 3 × 3 kernel) and projection (using a 1 × 1 kernel). Parametrized ReLU [84] was used to promote gradient backpropagation during training, resulting in better fine-tuning of the network hyperparameters. Additionally, images were augmented by using the CLoDSA image augmentation library [85], and the class imbalance problem was solved by using the SMOTE technique [86].

**Table 5 sensors-23-03618-t005:** Summary of papers on chronic wound detection.

Reference	Subject and Classes (Each Bullet Represents a Different Model)	Methodology	Original Images (Training Samples)	Results
Goyal et al. (2018) [65]	Diabetic foot ulcer Wound/Non-Wound	Cropping and resizing Augmentation was applied Faster R-CNN based on InceptionV2 was used for object detection	Normal: 2028 (28,392)Abnormal: 2080 (29,120)	mAP: 0.918
Amin et al. (2020) [42]	Diabetic foot ulcer Ischemia/Non-Ischemia Infection/Non-Infection	Objects were detected with the YOLOv2 algorithm. Detected objects were classified with a 16-layer CNN.	Ischemia: 9870 (9870) Infection: 5892 (5892)	Ischemia: Accuracy: 0.97 mAP: 0.95 Infection: Accuracy: 0.99 mAP: 0.9
Han et al. (2020) [87]	Diabetic foot ulcer Wagner Grades 0–5	K-means++ was used to select anchor box sizes based on the training dataset Fast R-CNN was used for detection and classification, and selected anchor boxes were used for region proposals	2688 (2668)	mAP: 0.9136
Anisuzzaman et al. (2022) [25]	Wound Wound/Non-Wound	Augmentation was applied YOLOv3 was used for wound detection	1800 (9580)	F1: 0.949 mAP: 0.973
Huang et al. (2022) [88]	Wound Blocked Blood Vessel/Suture/Ulceration	Augmentation was applied Fast R-CNN using the ResNet101 backbone was used for wound localization and classification The GrabCut and SURF algorithms were applied for wound boundary refinement	727 (3600)	Accuracy: 0.88 mAP: 0.87

The articles in which object detection was the main objective were filtered to obtain those with an mAP of at least 0.9 and a training dataset that contained 9000 or more samples (at least 1000 non-augmented samples); three articles were found. Goyal et al. analyzed multiple SOTA models that could be used in real-time mobile applications [65]. Faster R-CNN [88] with the InceptionV2 [89] backbone was found to be the best compromise among inference time (48 ms), model size (52.2 MB), and ulcer detection precision (mAP: 0.918). Two-step transfer learning was used to cope with the problem of small datasets. The first step partially transferred features from a classifier trained on the ImageNet [5] dataset, and the second step fully transferred features from a classifier trained on the MS COCO object detection dataset [76]. In another study, Amin et al. [42] proposed a two-step method for DFU detection and classification. First, an image was classified by using a custom shallow classifier (four convolutional layers) into the ischemic or infectious classes (accuracies: ischemia—0.976, infection—0.943); then, an exact DFU bounding box was detected by using YOLOv2 [90] with the ShuffleNet [91] backbone (average mAP—0.94). YOLOv2 with the ShuffleNet backbone outperformed conventional YOLOv2 by 0.22 mAP points when trained from scratch. The best mAP score (0.973) in DFU wound detection was achieved by Anisuzzaman et al. [25] with the next-generation YOLOv3 algorithm [92]. It was reported that the proposed model could process video taken by a mobile phone at 20 frames per second.

**Table 6 sensors-23-03618-t006:** Summary of studies on the semantic segmentation of chronic wounds.

Reference	Subject and Classes (Each Bullet Represents a Different Model)	Methodology	Original Images (Training Samples)	Results
García-Zapirain et al. (2018) [26]	Pressure ulcer Granulation/Slough/Necrotic	A CNN with two parallel branches with four convolutional layers was used to extract ROIs from HSI color space and Gaussian-smoothed images Morphological filters were applied ROIs was further processed by a CNN with four branches with eight convolutional layers; a prior model and LCDG modalities were used to make up the four input modalities	193 (193)	F1: 0.92
Li et al. (2018) [22]	Wound Wound/Non-Wound	Manual skin proportion evaluation YCbCr color space; the Cr channel was used to segment skin regions Images were normalized Augmentation was applied The 13 layers taken from MobileNet were used for skin region segmentation Semantic correction was applied Ambiguous background removal	950 (57,000)	IoU: 0.8588
Zahia et al. (2018) [36]	Pressure ulcer Granulation/Slough/Eschar	Flashlight reflection removal Image cropping to 5 × 5 patches Patch classification was performed with a CNN with three convolutional layers Image quantification was performed by using classified patches	Granulation: 22 (270,762) Necrotic: 22 (37,146) Slough: 22 (80,636)	Granulation: Accuracy: 0.9201 F1: 0.9731 Necrotic: Accuracy: 0.9201 F1: 0.9659 Slough: Accuracy: 0.9201 F1: 0.779
Jiao et al. (2019) [93]	Burn wound Superficial/Superficial Thickness/Deep Partial Thickness/Full-Thickness	Semantic segmentation was performed by using Mark R-CNN with the ResNet101FA backbone	1150	F1: 0.8451
Khalil et al. (2019) [16]	Wound Necrotic/Granulation/Slough/Epithelial	Image preprocessing was performed with CLAHE Statistical methods were used for color and texture feature generation NMF was used for feature map reduction GBT was used for pixel semantic segmentation	377	Accuracy: 0.96 F1: 0.9225
Li et al. (2019) [23]	Wound Wound/Non-Wound	Pixel locations were encoded Images were concatenated with location maps A MobileNet backbone was used for feature extraction A depth-wise convolutional layer was used to eliminate tiny wounds and holes	950	IoU: 0.86468
Rajathi et al. (2019) [57]	Varicose ulcer Granulation/Slough/Necrosis/Epithelial	Flashlight reflection removal Active contour segmentation was used for image segmentation Wound segments were cropped into 5 × 5 patches A CNN with two convolutional layers was used for patch classification Image quantification was performed by using classified patches	1250	Accuracy: 0.9955
Şevik et al. (2019) [94]	Burn wound Skin/Wound/Background	Images were split into 64 × 64 patches SegNet was used for semantic patch segmentation	105	F1: 0.805
Blanco et al. (2020) [59]	Dermatological ulcer Granulation/Fibrin/Necrosis/Non-Wound	Augmentation was applied Images were segmented by using super-pixels Super-pixels were classified by using the ResNet architecture Classified super-pixels were used for wound quantification	217 (179,572)	F1: 0.971
Chino et al. (2020) [66]	Wound Wound/Non-Wound	Augmentation was applied UNet was used for semantic segmentation (backbone not specified) Manual input for pixel density estimation Wound measurements were returned	446 (1784)	F1: 90
Muñoz et al. (2020) [95]	Diabetic foot ulcer Wound/Non-Wound	Semantic segmentation was performed by using Mark R-CNN	520	F1: 0.964
Wagh et al. (2020) [55]	Wound Wound/Skin/Background	Probabilistic image augmentation was applied DeepLabV3 based on ResNet101 was used for segmentation CRF was applied to improve segmentation accuracy	1442	F1: 0.8554
Wang et al. (2020) [32]	Foot ulcer Wound/Non-Wound	Cropping and zero padding Augmentation was applied The MobileNetV2 architecture with the VGG16 backbone was used to perform semantic segmentation. Connected component labeling was applied to close small holes.	1109 (4050)	F1: 0.9405
Zahia et al. (2020) [28]	Pressure ulcer Wound/Non-Wound	Mark R-CNN with the ResNet50 backbone was used for image semantic segmentation A 3D mesh was used to generate the wound top view and a face index matrix Matching blocks were used for wound pose correction Face indices and wounds with corrected poses were used to measure wound parameters	210	F1: 0.83
Chang et al. (2021) [96]	Burn wound Superficial/Deep Partial/Full Thickness	Mask R-CNN based on ResNet101 was used for semantic segmentation	2591	Accuracy: 0.913 F1: 0.9496
Chauhan et al. (2021) [64]	Burn wound Wound/Non-Wound	Augmentation was applied ResNet101 was used for feature extraction A custom encoder created two new feature vectors using different stages of ResNet101 feature maps A decoder with two convolutional layers, upsampling, and translation returned the final predictions.	449	Accuracy: 0.9336 F1: 0.8142 MCC: 0.7757
Dai et al. (2021) [68]	Burn wound Wound/Non-Wound	Style-GAN was used to generate synthetic wound images CASC was used to fuse synthetic wound images with human skin textures Burn-CNN was used for semantic segmentation	1150	F1: 0.893
Liu et al. (2021) [31]	Burn wound Superficial/Superficial Partial Thickness/Deep Partial Thickness/Full Thickness/Undebrided Burn/Background	Augmentation was applied HRNetV2 was used as an encoder, and one convolutional layer was used as a decoder	1200	F1: 0.917
Pabitha et al. (2021) [56]	Burn wound Superficial Dermal/Deep Dermal/Full-Thickness	Image noise removal Customized Mask R-CNN based on the ResNet backbone was used for semantic segmentation	1800	Segmentation: Accuracy: 0.8663 F1: 0.869 Classification: Accuracy: 0.8663 F1: 0.8594
Sarp et al. (2021) [49]	Wound Wound Border/Granulation/Slough/Necrotic	cGAN (UNet was used for resolution enhancement in the generator) was used for semantic segmentation	13,000	F1: 0.93
Cao et al. (2022) [18]	Diabetic foot ulcer Wagner Grades 0–5	Mask R-CNN based on ResNet101 was used for semantic segmentation	1426	Accuracy: 0.9842 F1: 0.7696 mAP: 0.857
Chang et al. (2022) [60]	Pressure ulcer Wound/Re-Epithelization Granulation/Slough/Eschar	SLIC was used for tissue labeling Augmentation was applied DeepLabV3 based on ResNet101 was used for both segmentation tasks	Wound And Reepithelization: 755 (2893) Tissue: 755 (2836)	Wound And Reepithelization: Accuracy: 0.9925 F1: 0.9887 Tissue: Accuracy: 0.9957 F1: 0.9915
Chang et al. (2022) [50]	Burn wound Wound/Non-Wound/Deep Wound	Augmentation was applied DeeplavV3+ algorithm with the ResNet101 backbone was used.	4991	Accuracy: 0.9888 F1: 0.9018
Lien et al. (2022) [58]	Diabetic foot ulcer Granulation Tissue/Non-Granulation Tissue/Non-Wound	Generation of 32 × 32 image patches RGB and gray-channel 32 × 32 patches were classified with ResNet18 Classified image patches were used for wound quantification	219	Precision: 0.91
Ramachandram et al. (2022) [48]	Wound Epithelial/Granulation/Slough/Necrotic	UNet based on a non-conventional backbone was used for wound area segmentation Wound areas were cropped and resized An encoder–decoder with the EfficientNetB0 encoder and a simplified decoder was used for wound semantic segmentation	Wound: 465,187Tissue: 17,000	Wound: F1: 0.61 Tissue: F1: 0.61
Scebba et al. (2022) [27]	Wound Wound/Non-Wound	Wound areas were detected by using MobileNet CNN. UNet was used for binary segmentation of detected areas.	1330	MCC: 0.85

The articles on semantic segmentation were filtered to obtain those with an F1 score of at least 0.89 and a training dataset size of 2000 or more samples (at least 1000 non-augmented samples); in total, five articles were found. Dai et al. proposed a framework [68] for synthetic burn image generation by using Style-GAN [67] and used those images to improve the performance of Burn-CNN [93]. The proposed method achieved an F1 score of 0.89 when testing on non-synthetic images. The authors used 3D whole-body skin reconstruction. Therefore, the proposed method can be used for the calculation of the %TBSA (total body surface area). It was stated that this framework could also be used to synthesize other wound types. Three different DeeplabV3+ [97] models with the ResNet101 [77] backbone were used to estimate burn %TBSA while using the rule of palms [50]. Two models were used for wound and palm segmentation, and the third was used for deep burn segmentation. The SLIC [98] super-pixel technology was used for accurate deep burn annotation. Deep burn segmentation achieved an F1 score of 0.902 (total wound area: 0.894; palm area: 0.99). Sarp et al. achieved an F1 score of 0.93 [49] on simultaneous chronic wound border and tissue segmentation by using a conditional generative adversarial network (cGAN). Though the proposed model “is insensitive to colour changes and could identify the wound in a crowded environment” [49], it was demonstrated that the cGAN failed to converge; therefore, training should be closely monitored by evaluating the loss and validity of the generated image masks. Wang et al. used MobileNetV2 [99] for foot ulcer wound image segmentation [32] and achieved a Dice coefficient of 0.94. The model was pre-trained on the Pascal VOC segmentation dataset [100]. The connected component labeling (CCL) postprocessing technique was used to refine the segmentation results, and this contributed to a slight performance increase.

## 4. Limitations

The articles included in this review varied in their reporting quality. Therefore, extracting and evaluating data while using one set of rules is problematic. Section 2.4 describes the strategies that were used to cope with unreported information. Article selection was performed by the main author, and though clear eligibility criteria were used, this might have imposed some limitations. Because the main goal of this review was not performance evaluation, there was no requirement for the reporting quality or size (page count) of the articles. To avoid distorting the reported results, the authors of this review imposed some constrains, such as the limitations on performance and training sample counts in Section 3.6 and the outlier removal (based on median values) in Section 3.5.

## 5. Discussion and Conclusions

The articles that were found and analyzed for this review demonstrated the importance and popularity of DFU diagnosis and monitoring as a research subject. The application of deep learning methods for this computer vision problem was found to be the most universal and to deliver the best results, but lack of training data makes this challenging. The academic community has failed to compete with enterprises (Swift medical [48], eKare [49]) simply due to a lack of diabetic foot imagery. Though good results have been achieved, the generalization capabilities of some deep learning models developed to date raise doubts.

Zhang et al. found in their review that most of the time, only hundreds of samples were used [11]. Other reviews identified the lack of public datasets as the main challenge [7,8,12]. Therefore, this article had the goal of taking an inventory of available public datasets that could be used for the creation of diverse training dataset. Though this criterion is arbitrary and does not necessarily ensure adequate training capabilities or good model generalization, it was found that more than 40% of the reviewed articles used less than 1000 original images. Wang et al. used excessive augmentation to increase their dataset size by sixty times [22]. On the other hand, Zahia et al. proposed a segmentation system that used decisions from a 5 × 5 pixel patch classifier to create picture masks [36]. They had 22 original pressure ulcer images, which resulted in a training dataset of 270,762 image patches. These two examples are extreme, but they illustrate the importance of data availability well.

In a review performed by Zahia et al., data preprocessing was considered as a conventional step in a typical CNN application pipeline [8]. It is considered to significantly improve performance [12]. Models’ generalization capabilities can also be affected by different image-taking protocols or even just different cameras. This issue is especially relevant in the context of home-based model usage. Though models can be trained on a large and diverse dataset to mitigate this risk, special attention should be paid to preprocessing techniques. In the current review, more than 30% of the articles did not report the use of any image preprocessing techniques, and only one used four [17]. Though this was not usually the case, the use of different public dataset calls for an approach to image adjustment that can later be directly applied during inference under any environmental conditions.

Augmentation is considered a solution to the problem of small datasets [8,11]. It was found that more than one-third of the reviewed articles did not report augmentation. Others reported using augmentation, but it was not possible to account for the increase in the training dataset. This made it difficult to evaluate the relationship between model performance and training dataset size. Though most articles used data augmentation, the effects of augmentation on the results were rarely evaluated. There are two big branches of augmentation techniques: classical and deep-learning-based [61]. Of the reviewed articles, only one used deep-learning-based augmentation [68]. Though augmentation can increase the dataset size by sixty times, the acceptable level of augmentation should be investigated.

The backbone is the core part of a deep learning model, as it generalizes data and provides feature maps for downstream processing. Though ResNet architectures of various depths were used the most, the resulting model performance had the widest spread, indicating that there is not a straightforward application of pre-trained backbones. Less than one-third of papers reported using a custom backbone design. While some were simple and extracted features of 5 × 5 pixel image patches [36,57], others provided model scaling capabilities [44] by using a custom number of standardized blocks. Models with custom backbones demonstrated the best performance and the smallest spread in the results, but this is to be expected considering the effort put into their design and fine-tuning. It should be noted that past reviews (referenced in the Introduction section) did not analyze this aspect. Finally, this is all related to data availability, as having more data helps design more complex custom architectures that do not overfit; at the same time, having more data results in better off-the-shelf backbone adoption through transfer learning or even training from scratch.

The best-performing models for each objective were reviewed in an effort to objectively find a common trend and direction for further research. In classification, past reviews found that CNNs based on residual connection were used the most (DenseNet [8], ResNet [12]), but no conclusions were made on their performance. In the current review, the best results were achieved by using backbones based on parallel convolutions, Inception, and custom architectures. There were only a few articles that focused on object detection, but the best performance was achieved by using the YOLO architecture family, and it was found to be the most popular in past reviews [11,12]. Semantic segmentation was found to be performed the most by using the Mask R-CNN and DeepLabV3 architectures with ResNet101 for feature extraction. These two methods were also in the two best-performing models. Other reviews found Mask R-CNN [8], UNet [12], and other FCNs [11] to be used the most, but conclusions about their performance were not reported.

The observations in this review have indicated the following problems for future research:There are poor preprocessing strategies when images are taken from multiple different sources. Methods that can improve uniformity and eliminate artifacts introduced in an uncontrolled environment should be analyzed.There is a lack of or excessive use of augmentation techniques. Augmentation strategies that yield better model generalization capabilities should be tested. Deep-learning-based methods should be applied.There is a limited amount of annotated data. An unsupervised or weakly supervised deep learning model for semantic segmentation should be developed.

## Figures and Tables

**Figure 1 sensors-23-03618-f001:**
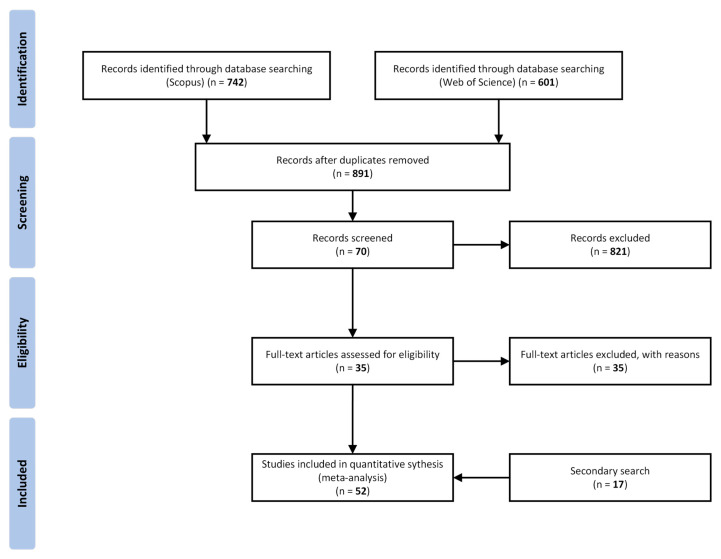
PRISMA flow diagram for article selection.

**Figure 2 sensors-23-03618-f002:**
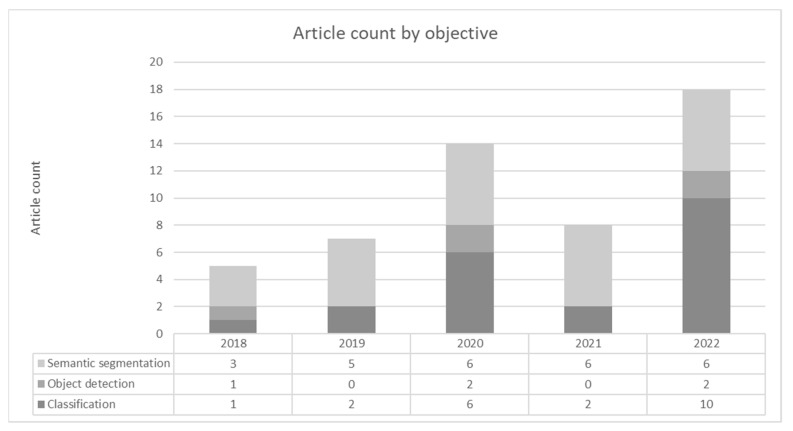
Distribution of search results by year and objective.

**Figure 3 sensors-23-03618-f003:**
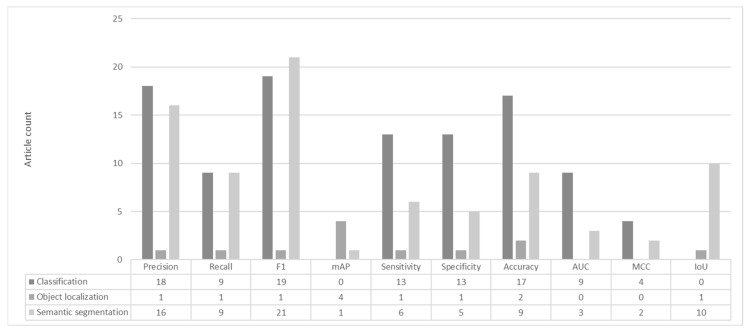
Usage counts of performance metrics.

**Figure 4 sensors-23-03618-f004:**
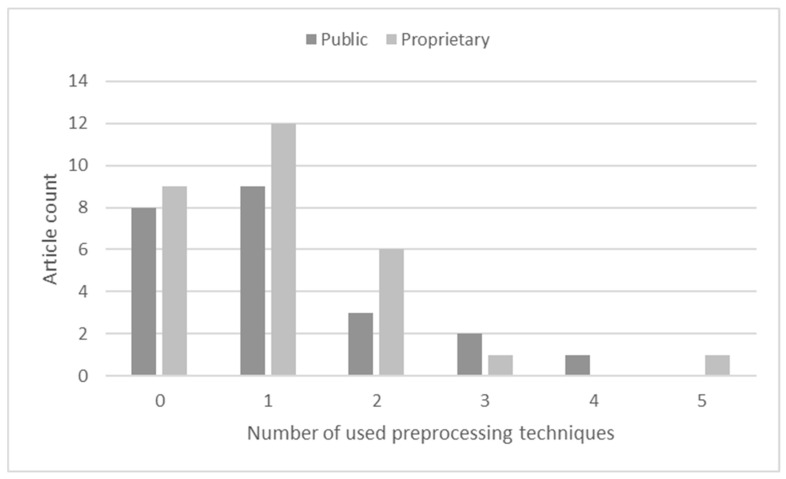
Preprocessing technique usage.

**Figure 5 sensors-23-03618-f005:**
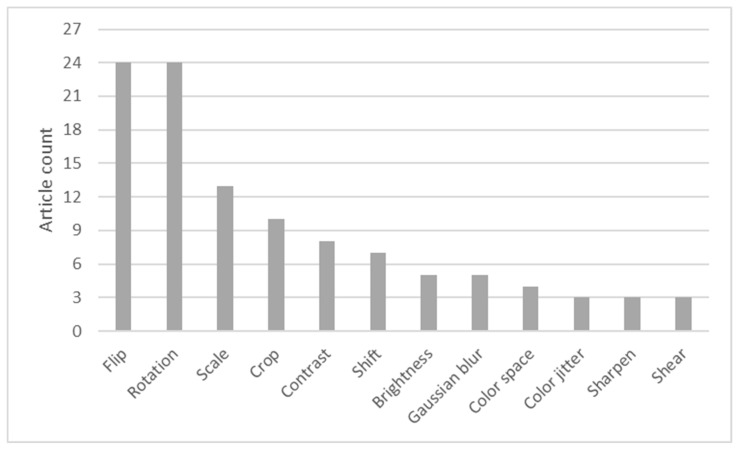
The image augmentation techniques used.

**Figure 6 sensors-23-03618-f006:**
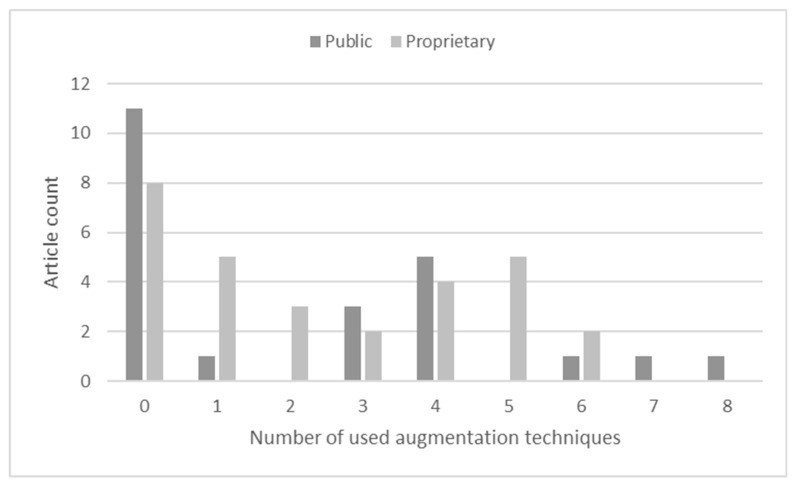
Number of augmentation techniques used.

**Figure 7 sensors-23-03618-f007:**
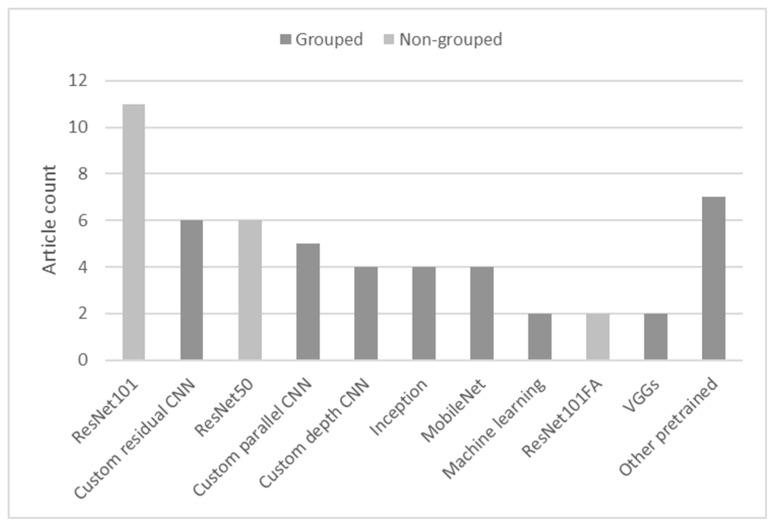
Backbone architecture usage.

**Figure 8 sensors-23-03618-f008:**
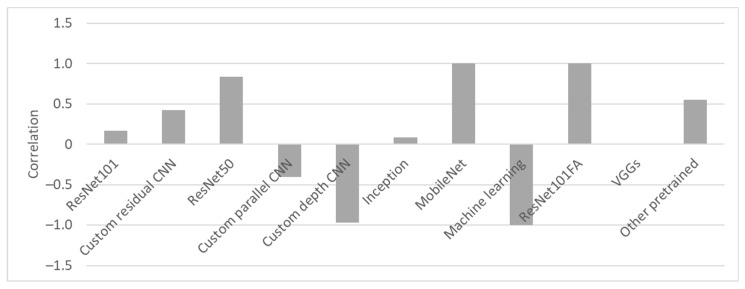
Correlation of the backbone performance with training dataset size.

**Figure 9 sensors-23-03618-f009:**
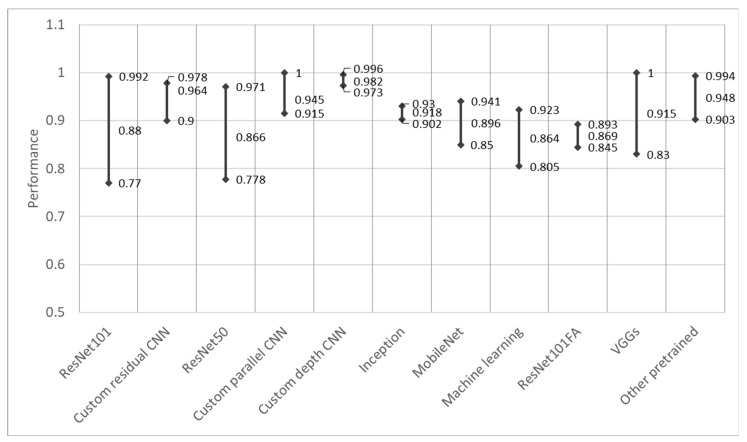
Backbone performance evaluation.

**Table 1 sensors-23-03618-t001:** Article eligibility criteria.

Inclusion Criteria	Exclusion Criteria
Published in English	
Journal articles	Conference proceedings, unpublished articles, and reviews
Full text available	Abstract or full text not available
Research subject of chronic wounds: Diabetic foot ulcer Pressure ulcer Varicose ulcer Other lower-extremity ulcers Burns	Other types of dermatological pathologies
Computer vision used for: Classification Object detection Semantic segmentation	Unrelated to computer vision
Red, green, and blue (RGB) images were used as the main modality	Thermograms, spectroscopy, depth, and other modalities requiring special equipment were used as the main modality
Deep learning technologies were used	Only machine learning or statistical methods were used
Performance was reported with any of the following metrics: Accuracy Sensitivity Specificity Precision Recall F1 (or DICE or F score) Area under the curve (AUC) Matthews correlation coefficient (MCC) Mean average precision (mAP) Intersection over union (IoU)	Performance was not reported

**Table 2 sensors-23-03618-t002:** Model performance metrics [15,16,17,18].

Metric	Formula		Description
Accuracy	$\frac{TP+TN}{TP+FP+TN+FN}$	$\left(2\right)$	Correct prediction ratio
Sensitivity	$sn=\frac{TP}{TP+FN}$	$\left(3\right)$	Fraction of correct positive predictions
Specificity	$sp=\frac{TN}{TN+FP}$	$\left(4\right)$	Fraction of correct negative predictions
Precision	$p=\frac{TP}{TP+FP}$	$\left(5\right)$	Correct positive predictions over all positive predictions
Recall	$\frac{TP}{TP+TN}$	$\left(6\right)$	Correct positive predictions over all correct predictions
F1 (or DICE or F score)	$\frac{2\cdot TP}{2\cdot TP+FP+FN}$	$\left(7\right)$	Harmonic mean between precision and recall
AUC (area under the curve)	$\frac{sn+sp}{2}$	$\left(8\right)$	Threshold-invariant prediction quality
MCC (Matthews correlation coefficient)	$\frac{TP\cdot TN-FP\cdot FN}{\sqrt{\left(TP+FP\right)\left(TP+FN\right)\left(TN+FP\right)\left(TN+FN\right)}}$	$\left(9\right)$	Correlation between prediction and ground truth
mAP (mean average precision)	$\frac{1}{classes}\underset{classes}{\sum }\bar{p}$	$\left(10\right)$	Average precision of all classes

**Table 3 sensors-23-03618-t003:** Publicly available datasets.

Nr.	References	Image Count	Sets Used	Wound Types	Annotation Types	Use Count	External Use References
1	[19]	188	1	- Chronic wounds	Mask	0	
2	[20]	74	1	- Burns		0	
3	[21]	4000	1	- Diabetic foot ulcers	Mask	0	
4	[22]	N/A	1	- Diabetic foot ulcers - Pressure ulcers - Burns		2	[23]
5	[24]	594	8	- Diabetic foot ulcers - Pressure ulcers - Burns - Venous or arterial leg ulcer s - Malignant wounds - Dehisced wounds resulting from surgical wound infection - Skin or microvascular changes associated with diabetes - Hemangioma		9	[16,22,23,25,26,27,28,29,30]
6	[31]	1200	1	- Burns		1	
7	[25,32,33,34]	3867	4	- Diabetic foot ulcers - Pressure ulcers - Venous leg ulcers - Surgical wounds	ROI Mask	4	
8	[35]	40	2	- Pressure ulcers		2	[16,36]
9	[37]	210	1	- Face wounds - Hand wounds - Back wounds - Foot wounds		1	[27]
10	[38,39,40]	5659	2	- Diabetic foot ulcers	Class labe lROI	10	[17,18,27,41,42,43,44,45,46]
11	[47]	1000	1	- Skin disease		1	
	Total:	11,173					

**Table 4 sensors-23-03618-t004:** Summary of studies on chronic wound classification.

Reference	Subject and Classes (Each Bullet Represents a Different Model)	Methodology	Original Images (Training Samples)	Results
Goyal et al. (2018) [51]	Diabetic foot ulcer Wound/Non-Wound	Cropping and resizing Augmentation was applied A 14-layer DFUNet CNN with parallel convolutions was used for binary classification	344 (22,605)	Accuracy: 0.896 F1: 0.915
Cirillo et al. (2019) [63]	Burn wound Superficial/Pre-Intermediate/Post-Intermediate/Full/Skin/Background	224 × 224 image patches were cropped Augmentation was applied ResNet101 was used for classification	23 (676)	Accuracy: 0.9054
Zhao et al. (2019) [53]	Diabetic foot ulcer Wound Depth (0–4) Granulation Tissue Amount (0–4)	256 × 256 image patches were cropped Sharpening was applied Augmentation was applied Bi-CNN was used for classification. VGG16 backbones were used for depth and granulation paths.	1639	Wound Depth: Accuracy: 0.8336 F1: 0.83024 Granulation Tissue Amount: Accuracy: 0.8334 F1: 0.82278
Abubakar et al. (2020) [52]	Burn wound Wound/Non-Wound	Cropping and normalization Augmentation was applied ResNet50 was used for image classification	1900	Accuracy: 0.964
Alzubaidi et al. (2020) [79]	Diabetic foot ulcer Wound/Non-Wound	Cropping and resizing Augmentation was applied A 58-layer DFU_QUTNet was used for feature extraction Extracted features were used in an SVM classifier	754 (20,917)	F1: 0.945
Alzubaidi et al. (2020) [47]	Diabetic foot ulcer Wound/Non-Wound	224 × 224 image patches were cropped A CNN was trained by using a dataset of the same domain A CNN with 29 convolutional layers (parallel and simple) was used for classification	1200 (2677)	F1: 0.976
Chauhan et al. (2020) [62]	Burn wound Severe/Low/Moderate	Augmentation was applied by using label-preserving transformations ResNet50 was used for body part classification Body-part-specific ResNet50 was used for feature extraction, and an SVM was used to classify burn severity	141 (316)	Accuracy: 0.8485 F1: 0.778
Goyal et al. (2020) [40]	Diabetic foot ulcer Ischemia/Non-Ischemia Infection/Non-Infection	Cropping and resizing Natural augmentation was applied InceptionV3, ResNet50, and InceptionResNetV2 were used for feature extraction An SVM classifier used extracted bottleneck features for classification	Ischemia: 1459 (9870) Infection: 1459 (5892)	Ischemia: Accuracy: 0.903 F1: 0.902 MCC: 0.807 Infection: Accuracy: 0.727 F1: 0.722 MCC: 0.454
Wang et al. (2020) [54]	Burn wound Shallow/Moderate/Deep	Images were cropped and resized Augmentation was applied ResNet50 was used for classification	484 (5637)	F1: 0.82
Rostami et al. (2021) [33]	Wound Background/Normal Skin/Various ulcer/Surgical Wound	Augmentation was applied The AlexNet classifier was used for whole-image classification 17 image patches were cropped The AlexNet classifier was used for patch classification An MLP classifier used the results of the two AlexNet classifiers to make a final decision	400 (19,040)	Accuracy: 0.964 F1: 0.9472
Xu et al. (2021) [45]	Diabetic foot ulcer Ischemia/Non-Ischemia Infection/Non-Infection	The DeiT vision transformer was used for feature extraction An MLP reduce feature map dimensionality for storage in a knowledge bank Classification was based on cosine similarity between test images and stored knowledge	Ischemia: 9870 (9870) Infection: 5892 (5892)	Ischemia: Accuracy: 0.909 F1: 0.903 Infection: Accuracy: 0.78 F1: 0.782
Al-Garaawi et al. (2022)	Diabetic foot ulcer Wound/Non-Wound Ischemia/Non-Ischemia Infection/Non-Infection	Augmentation was applied Local binary patterns were extracted and converted into 3D representations RGB and LBP were summed The sum was fed into a 12-layer CNN.	Wound: 1679 (16,790) Ischemia: 9870 (9870) Infection: 5892 (5892)	Wound: Accuracy: 0.93 F1: 0.942 Ischemia: Accuracy: 0.99 F1: 0.99 Infection: Accuracy: 0.742 F1: 0.744
Al-Garaawi et al. (2022) [43]	Diabetic foot ulcer Wound/Non-Wound	GLCM, Hu moment, color histogram, SURF, and HOG feature extraction A CNN with four residual blocks was used for feature extraction A logistic regression classifier was used to classify images	1679 (1679)	Accuracy: 0.9424 F1: 0.9537
Alzubaidi et al. (2022) [30]	Diabetic foot ulcer Wound/Non-Wound	Cropping and resizing Augmentation was applied A CNN with 29 convolutional layers (parallel and simple) was used for classification	3288 (59,184)	F1: 0.973
Anisuzzaman et al. (2022) [29]	Wound Background/Normal Skin/Various ulcers/Surgical Wound	Images were cropped and locations were labeled Data augmentation was applied The VGG19 classifier was used for wound image classification An MLP was used for wound location classification The results of previous classifiers were concatenated An MLP was used for the final classification	1088 (6108)	F1: 1
Das et al. (2022) [80]	Diabetic foot ulcer Wound/Non-Wound	Cropping and resizing A CNN with three parallel convolutional layers was used for classification	397 (3222)	Accuracy: 0.964 F1: 0.954
Das et al. (2022) [81]	Diabetic foot ulcer Wound/Non-Wound Ischemia/Non-Ischemia Infection/Non-Infection	Grayscale image conversion Gabor and HOG feature extraction GoogLeNet was used to extract features A random forest classifier was used to classify images	Wound: 1679 (1679) Ischemia: 9870 (9870) Infection: 5892 (5892)	Wound: Accuracy: 0.88 F1: 0.89 Ischemia: Accuracy: 0.92 F1: 0.93 Infection: Accuracy: 0.73 F1: 0.76
Das et al. (2022) [44]	Diabetic foot ulcer Ischemia/Non-Ischemia Infection/Non-Infection	Res4Net (network with four residual blocks) was used for ischemia classification Res7Net (network with seven residual blocks) was used for infection classification	Ischemia: 9870 (9870) Infection: 5892 (5892)	Ischemia: Accuracy: 0.978 F1: 0.978 MCC: 0.956 Infection: Accuracy: 0.8 F1: 0.798 MCC: 0.604
Liu et al. (2022) [41]	Diabetic foot ulcer Ischemia/Non-Ischemia Infection/Non-Infection	Augmentation was applied EfficientNet networks were used for classification (B1 for ischemia, B5 for infection)	Ischemia: 2946 (58,200) Infection: 2946 (58,200)	Ischemia: Accuracy: 0.9939 F1: 0.9939 Infection: Accuracy: 0.9792 F1: 0.9792
Venkatesan et al. (2022) [78]	Diabetic foot ulcer Wound/Non-Wound	Natural dataset transformation using CLoDSA SMOTE was applied for class balancing A 22 layer CNN with parallel convolutions was used for classification	1679 (18,462)	Accuracy: 1 F1: 1 MCC: 1
Yogapriya et al. (2022) [17]	Diabetic foot ulcer Infection/Non-Infection	Augmentation was applied A custom CNN DFINET is used for image classification	5892 (29,450)	Accuracy: 0.9198 F1: 0.9212 MCC: 0.84

## Data Availability

Not applicable.

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
