# Peer review of "Towards Home-Based Diabetic Foot Ulcer Monitoring: A Systematic Review"

_sensors, 2023, doi:10.3390/s23073618_

Round 1

Reviewer 1 Report

This paper studied articles about the diabetic foot ulcer area detection and segmentation by image analysis based on the deep learning technology. Although some contributions were presented, few innovations were found.

1 The article search and selection process is based on PRISMA methodology which derived from Ref[13].

2 Although a lot of analysis were made in Section 3, the work seems to focus on the data statistics instead of the technology analysis. Some analyses can be completed easily by using developed programs. The latest research results about the diabetic foot ulcer area detection and segmentation are expected to be concluded.

3 The abstract is not appropriate since the background were introduced mainly. The abstract is recommended to summarize the study methods, procedures, and highlight the contributions.

Author Response

Please find a response in attached file.

Reviewer 2 Report

The article "Towards home-based diabetic foot ulcer monitoring. Systematic review" is well-written and, from my point of view, would be of interest to the readers of MDPI Sensors. However, despite these, and before its publication, I would recommend the following:

Although the authors listed the paper's contributions in the introduction, I don't see the academic contribution. The article looks more like a pure literature review, where the opening and the references shown do not support the conclusion. Authors should compare their contributions with similar works in the same domain.

Author Response

Thank you for reviewing our manuscript. Attached are answers to your remarks.

Reviewer 3 Report

Thank you for the chance to review the present manuscript. The authors provide a well-performed systematic review of home-based diabetic foot ulcer monitoring. I have some minor comments I recommend addressing before publication.

1)the introduction section mentioned multimodal data processing in different ways, which might be confusing for readers. For example, the authors write:

-Chan and Lo [9] have performed a review of existing DFU monitoring systems, some using multiple modalities“

-„Anisuzzaman et al. [10] concludes that multimodal datasets are needed to apply Artificial Intelligence (AI) methods for clinical wound analysis“

-„Multiple past reviews have concluded that multimodal data is able to deliver better classification, detection and semantic segmentation results in wound clinical care“

However, Chan et al. refer to different image modality techniques and Anisuzzaman to multimodal datatypes (e.g., wound images, wound-specific electronic health records). I recommend adding a section regarding multimodal datatypes and their usefulness in automatized DFU diagnosis. Here, you can also mention hybrid models, which can process multiple datatypes in one model (e.g., see: https://doi.org/10.1038/s41591-022-01981-2 or DOI: 10.3390/jpm12040509)

1)Table 1 exclusion criteria „etc.“ should be removed and/or specified. „etc.“ will not help to repeat the methodology.

2) why did you limit the search to articles published since 2018 and not consider studies published before? This should be justified in the paper or added to the limitation section.

3)the intersection over union (IoU) metric provided in figure 2 is not listed in Table 1 as an inclusion criteria metric.

4)please also add to the limitation section that only one reviewer performed the screening and data extraction, whereas common guidelines on systematic reviews recommend at least two reviewers working independently to prevent missing information (e.g., see here: DOI: 10.1002/jrsm.1369)

Author Response

(The authors gave the same response as above.)

Reviewer 4 Report

The level of originality of the paper is high. The literature review and proposed methodology of Towards home-based diabetic foot ulcer monitoring are properly discussed and compared to the previous studies.

In this paper, authors used many sources, containing both historical and fundamental works, as well as the latest scientific research on this topic. But the literature review can be structured. The papers discussed many points of this study. Please, discuss the newest papers:

QoS-Ledger: Smart Contracts and Metaheuristic for Secure Quality-of-Service and Cost-Efficient Scheduling of Medical-Data Processing. Electronics, 10, 3083. https://doi.org/10.3390/electronics10243083

Development of Polymer Film Coatings with High Adhesion to Steel Alloys and High Wear Resistance. Polymer Composites, 41(7), 2875-2880. https://doi.org/10.1002/pc.25583

Emotional Development in Preschoolers and Socialization. Early child development and care, 191, 16. https://doi.org/10.1080/03004430.2020.1717480

The introduction section has benefit from having a clearer structure of what to expect in the paper. Furthermore, the author(s) would benefit from being more concise in their writing, as much of the content was redundant and overemphasized. While it is good practice to assume the reader has no prior knowledge of the content, a topic and/or discussion does not need to be explained over and over again if it is stated both adequately and appropriately once.

Some conclusions contribute to the study of the problem. The author does not formulate the problem itself – it makes impossible to analyse the contribution of the paper. The aim or the question of the paper (or even the hypothesis of the author) are formulated.

Overall, it is very clear to grasp understanding of the manuscript and content in its current state. I strongly advise using hypothesis points to articulate and/or express material in scientific writing. Publication of this piece seems likely in any reputable scientific periodical after a correction in the writing of the manuscript.

Tables are important to explore the specifics. But should add some conclusions about every table into text to the study of the problem.

Authors need to add more details on the range of simulation considered in this work should be clearly outlined within the abstract. The current statements are vague and too general to get an idea of the work that have been accomplished.

The paper possesses a proper form of well-structured and readable technical language of the field and represents the expected knowledge of the journal`s readership.

There are minor errors in English, but this does not affect the general nature of the work. The current study brings many new to the existing literature or field. For one, the author(s) seem to have a good grasp of the current literature on their topic area (i.e., recent literature and seminal texts relevant to their study is not cited/referenced).

Author Response

(The authors gave the same response as above.)

Round 2

Reviewer 1 Report

In the responses authors stated that “Article aims to investigate if dataset creation was properly facilitated through public dataset usage, pre-processing and augmentation as well as soundness of reported solutions”. Therefore, authors should focus on the training data. The detailed introduction of the article search and selection process in Section 2 seems unnecessary. Sections 2.1,2.2 and 2.3 are recommended to be introduced briefly.

Author Response

Thank you for constructive review. Please find our response in a file.
